# Correlating Basal Gene Expression across Chemical Sensitivity Data to Screen for Novel Synergistic Interactors of HDAC Inhibitors in Pancreatic Carcinoma

**DOI:** 10.3390/ph16020294

**Published:** 2023-02-14

**Authors:** Nemanja Djokovic, Ana Djuric, Dusan Ruzic, Tatjana Srdic-Rajic, Katarina Nikolic

**Affiliations:** 1Department of Pharmaceutical Chemistry, Faculty of Pharmacy, University of Belgrade, Vojvode Stepe 450, 11221 Belgrade, Serbia; 2Department of Experimental Oncology, Institute for Oncology and Radiology of Serbia, Pasterova 14, 11000 Belgrade, Serbia

**Keywords:** bioinformatics, PDAC, synergism, sphingolipid signaling, HDAC inhibitors, ROCK inhibitors

## Abstract

Pancreatic ductal adenocarcinoma (PDAC) is one of the most aggressive and lethal malignancies. Development of the chemoresistance in the PDAC is one of the key contributors to the poor survival outcomes and the major reason for urgent development of novel pharmacological approaches in a treatment of PDAC. Systematically tailored combination therapy holds the promise for advancing the treatment of PDAC. However, the number of possible combinations of pharmacological agents is too large to be explored experimentally. In respect to the many epigenetic alterations in PDAC, epigenetic drugs including histone deacetylase inhibitors (HDACi) could be seen as the game changers especially in combined therapy settings. In this work, we explored a possibility of using drug-sensitivity data together with the basal gene expression of pancreatic cell lines to predict combinatorial options available for HDACi. Developed bioinformatics screening protocol for predictions of synergistic drug combinations in PDAC identified the sphingolipid signaling pathway with associated downstream effectors as a promising novel targets for future development of multi-target therapeutics or combined therapy with HDACi. Through the experimental validation, we have characterized novel synergism between HDACi and a Rho-associated protein kinase (ROCK) inhibitor RKI-1447, and between HDACi and a sphingosine 1-phosphate (S1P) receptor agonist fingolimod.

## 1. Introduction

Pancreatic ductal adenocarcinoma (PDAC) is considered as one of the most aggressive and lethal malignancies. Early onset dedifferentiation and metastasis significantly humper early diagnosis and treatment options which contributes to the devastatingly poor prognosis of the PDAC [1,2,3]. Although the surgical resection followed by adjuvant chemotherapy offers some prospects of long-term survival, over 80% of PDAC patients are diagnosed in terminal stages when distant metastases have already occurred leaving the chemotherapy as the most frequently applied treatment option [4,5,6]. Additionally, the most of the patients who have received surgery suffer from recurrence within a year, which makes the chemotherapy the mainstay of PDAC therapy [6]. Considering the very modest improvement in 5-years survival rate since 1970s from 3% to 11%, there is an urgent need to develop non-surgical therapeutic approaches for effective treatment of pancreatic cancer [7,8].

Gemcitabine, with or without additional chemotherapeutics, is considered as a first line option in chemotherapy of PDAC. Being one of the most chemoresistant cancers, efficacy of currently available chemotherapeutics for PDAC is seriously compromised by a promptly developing chemoresistence [2,6]. Dense stromal environment together with many genetic and epigenetic alterations contribute to the complex mechanisms of the resistance. Inspired by the many epigenetic alterations in the PDAC, preclinical studies investigating the interference of epigenetic therapeutics with epigenetic mechanisms underlying the PDAC resulted in the promising findings. However, increasing amount of evidence suggests that the epigenetic therapeutics offer measurable benefits in PDAC only in combined therapy settings [5,8]. Histone deacetylase inhibitors (HDACi) represent some of the most promising epigenetic therapeutics for the treatment of PDAC [8]. Currently, there are four HDACi approved by US Food and Drug Administration (FDA) while many candidate drugs are in the clinical trials. Despite the many promising preclinical results, the clinical trials investigating the efficacy of HDACi as a single agent in solid tumors gave rather disappointing results [9]. Therefore, the combination of these potent anticancer agents with other chemotherapeutics is proposed to be one of the most promising approaches in the future development of HDACi-based chemotherapy [8,10].

Despite significant breakthroughs in the cancer research during the last decades, high mortality remains a major issue which highlights the urgent need for novel therapeutic approaches to the PDAC. A combination therapy holds the promise for advancing the treatment of PDAC due to the possibility to reduce the development of drug resistance. However, in experimental settings, the number of possible combinations considering the all approved drugs and drug candidates is too large and the success rates of such experiments are low. In an emerging era of personalized medicine, bioinformatics approaches could offer more rational way to select the optimal combination of drugs for the specific sub-population of patients. In this work, we explored possibility of using drug-sensitivity data together with a basal gene expression data on pancreatic cell lines to predict the combinatorial options available for HDACi. Our results identified the sphingolipid signaling pathway with associated downstream effectors as a promising targets for combined therapy with HDACi or future development of multi-target therapeutics. Through the process of experimental validation of the results, novel synergisms between HDACi and a Rho-associated protein kinase (ROCK) inhibitor RKI-1447, and between HDACi and a sphingosine 1-phosphate (S1P) receptor agonist fingolimod have been characterized.

## 2. Results and Discussion

Pharmaco-transcriptomics relationships between basal gene expression and chemical sensitivity of pancreatic carcinoma cell lines (CCLs) were investigated using the systematic correlation analysis of large publically available data sets. Correlating the chemical sensitivity data to the basal gene expression was previously shown to be the valuable method for revealing mechanisms of action of small molecules [11], or in defining the therapy response signatures for limited number of HDACi [12]. In this study, the Genomics of Drug Sensitivity in Cancer (GDSC) [13] and the Cancer Therapeutics Response Portal (CTRP) [11] chemical sensitivity data sets for 864 small molecules across up to 38 pancreatic CCL were correlated to the basal gene expression of 19,221 genes obtained from the Cancer Cell Line Encyclopedia (CCLE) [14]. Pearson correlation coefficients have been calculated between area under the curve (AUC) values and basal expression data (expressed as log2RSEM values) for each transcript. Matrix of Pearson correlation coefficients was further normalized using Fisher’s *z*-transformation to adjust for variations in number of tested CCLs for each of analyzed molecules. Generally speaking, transcripts negatively correlated with the chemical sensitivity data could be attributed to the sensitivity of the pancreatic CCLs towards corresponding small molecule, while the positively correlated transcripts could be attributed to the resistance of the CCLs. In alignment to this, transcripts corresponding to the Tukey’s outliers of each small molecule were further extracted and stored as a transcriptomics “fingerprints” of drug sensitivity or resistance. Stored transcriptomics “fingerprints” were further used for predictions of possible novel small molecule combinations. Under assumptions that genes from resistance fingerprints are directly involved in a cancer cell machinery for escaping the drug-induced cell death, while genes from sensitivity fingerprints are directly involved in the mechanisms which sensitize cancer cells on therapy, we hypothesized that level of overlap between these fingerprints could be an indicator of synergism between small molecules and could identify novel pharmacological targets for dual targeting in PDAC. In other words, synergistic effects between two pharmacological agents are hypothesized to occur when first pharmacological agent which is able to sensitize the cancer cells utilizing the same, or similar set of genes as the ones involved in developing of resistance on the second pharmacological agent. The general approach of the study is presented on the Figure 1.

To further explore this hypothesis and identify novel potential synergistic counterparts to the HDACi, counting of the transcripts’ overlaps was performed on the pool of correlation data where only transcripts with significant correlation were considered (see Materials and Methods). Namely, if the transcripts with significant positive correlation of the first small molecule response had overlapped with the transcripts related to negative correlations of the second small molecule response, these associations were counted and stored. Scores for all possible combinations of HDACi and the rest of small molecules were calculated and the probability of each of combination was evaluated as a count of all possible overlaps between significantly correlated transcripts (named “Final_Syn_Score” in the Appendix A). Final results included 29,233 possible combinations between HDACi and small molecule interactors (see Appendix A).

Using the proposed protocol, several known synergistic interactors of HDACi have been identified which added up to the validity of the approach. For example, the synergistic interaction between DNMT1 inhibitor and pan-HDACi was recently described on MIA PaCa2 cells [15]. Sorafenib is one of the kinase inhibitors (BRAF; FLT3; KDR;RAF1) with preclinical data on synergism with HDACi and this combination is currently under clinical investigations for usage in PDAC [8,16]. Another positive examples include combination of HDAC inhibitors with PI3K inhibitor [17], mTOR inhibitor [18], BET (BRD4 in Table 1) inhibitors [19], and combination with gemcitabine [10] (Table 1). However, it should be noted that not all of the inhibitors of abovementioned pharmacological targets were highly scored in our analysis (for full table of predictions see Appendix A) which could be attributed to the mostly unknown off-target effects of the small molecules.

In order to identify the possible signaling pathways which could be targeted to achieve the synergistic effects with HDACi in the PDAC, a pathway enrichment analysis of annotated targets for predicted small molecule counterparts of HDACi was performed. Enrichment analysis revealed sphingolipid signaling pathway (as defined in KEGG database) [20,21] as one of the top-scored pathways involved in the predicted synergies across analyzed datasets of small molecules (Figure 1 and Appendix A). Sphingolipid signaling pathway (as defined through the analyzed KEGG and Wiki Pathway databases [22]) encompass several enzymes involved in the sphingolipid metabolism (Sphingosine Kinase (SPHK), Ceramide Kinase (CERK), Ceramide Synthase (CERS), Alkaline Ceramidase (ACER), Acid Ceramidase (ASAH) etc.) as well as sphingosine-1-phosphate membrane receptors (S1PR1-5) and their downstream Rho/ROCK, PI3K/Akt and MAPK pathways. It is interesting to note that current literature data do not recognize any of the elements of sphingolipid signaling (except downstream pathways activated by diverse signals) as potential novel therapeutic targets for achieving the synergism with HDACi.

One of the central pillars of the sphingolipid signaling is so-called sphingolipid rheostat [23]. Sphingolipid rheostat, as a concept within sphingolipid metabolism, could be seen as a dynamical equilibrium between amounts of pro-apoptotic ceramide and mitogenic and anti-apoptotic sphingosine 1-phosphate (S1P). Sphingolipid rheostat, and particularly equilibrium between ceramide and S1P, have been just recently characterized as one of the critical regulators of the pro- and anti-apoptotic signaling in a metabolically dynamic pancreatic cancers [24,25]. Additionally, Speirs et al. identified SPHK1 as a key driver of the conserved S1P: Ceramide imbalance in the pancreatic cancer subcultures and therefore one of the central controlling hubs of the rheostat machinery [24]. The most recent literature data suggested that sphingolipid signaling pathway could be an important novel therapeutic target to suppress the proliferation across pancreatic tumors made up of heterogeneous cell populations, with recognition of potential of SPHK1 inhibitors as an approach to reverse healthy balance of pro- and anti-apoptotic signaling in pancreatic cancers [24,26,27,28]. In addition to this, our predictions identified synergism between pan-HDAC inhibitor and SPHK1 inhibitor to be highly scored (in first 5% of the ordered list of predictions) (Table 1 and Appendix A).

To further corroborate our results on potential synergism occurring between HDACi and sphingolipid signaling pathway in pancreatic carcinoma, expression analysis was performed on the data obtained from pancreatic cancer cohort (The Cancer Genome Atlas (TCGA) database—number of samples 179) and healthy control (Genotype-Tissue Expression (GTEx) database—number of sample 171) (Figure 2). Besides overexpression of class I HDACs in pancreatic carcinoma patients, results indicated significant elevation of some of the key elements of sphingolipid signaling, including the enzymes involved in sphingolipid rheostat. Namely, expression of SPHK1, but not SPHK2, in patient tissue was significantly increased. Additionally, according to the differential expression analysis (Figure 2), expressions of many components of the sphingolipid signaling pathway were significantly perturbed in a pancreatic cancer tissue, with most of them being upregulated. In summary, the expression analysis suggested significant perturbations of the sphingolipid signaling pathway in the pancreatic carcinoma. Considering the overexpression of HDACs in the pancreatic carcinoma and synergy predictions, these findings further supports exploration of dual targeting of HDACs and sphingolipid signaling as novel approach in treatment of PDAC (Figure 2).

As a final step of our bioinformatics analysis, interaction network of all of the overlapped transcripts was constructed. Under assumption that these overlapped transcripts could be related to the mechanisms of synergy happening on the protein level, network was created utilizing the STRING database which integrates all known and predicted associations between proteins, including physical interactions and functional associations [29].

Aiming to find underlying mechanism of synergy, network was constructed using overlapped transcripts as nodes and STRING interaction annotations as edges. Betweenness Centrality (BC) measurements were performed on created network to identify the most important communication hubs inside constructed network, e.g., the nodes with the largest number of intersections of the shortest communication paths between other nodes. As a node with the highest BC value emerged p53 tumor suppressor, indicating importance of p53 in the networks mediating the synergistic effects between HDACi and modulators of sphingolipid signaling.

Through the complex interactions with HDACs including the post-translational regulation of acetylation status, p53 was found to be an important regulator of HDACi-mediated cancer cell death [30]. Additionally, literature data indicates that pro-apoptotic ceramide accumulation is an important downstream mediator of the p53 response further corroborating involvement of p53 in a mechanism of potential synergy between HDAC inhibition and modulation of sphingolipid signaling [31]. Another identified communication hubs with corresponding BC values are presented in the Table 2. Besides the BC, another classical centrality measures such as Degree and Closeness were considered in identification of influential nodes in the constructed network. Moreover, additional centrality measures unambiguously identified p53 as one of the most influential nodes (Appendix A).

To evaluate predictive power of proposed approach, synergisms between novel HDACi synthesized in our group with modulators of sphingolipid signaling available in our laboratory at the time of the study have been analyzed. For this analysis we have used a set of recently developed HDAC inhibitors: Compound 6b (IC_50_ (HDAC1) = 4.73 μM, IC_50_ (HDAC3) = 1.86 μM, IC_50_ (HDAC6) = 0.19 μM, IC_50_ (HDAC8) = 2.44 μM); Compound 8b (IC_50_ (HDAC1) = 0.41 μM, IC_50_ (HDAC3) = 0.21 μM, IC_50_ (HDAC6) = 0.12 μM, IC_50_ (HDAC8) = 0.24 μM); Compound 9b (IC_50_ (HDAC1) = 1.47 μM, IC_50_ (HDAC3) = 3.47 μM, IC_50_ (HDAC6) = 0.03 μM, IC_50_ (HDAC8) = 0.71 μM) [32]. 

One of the pairs listed among 2.3% of sorted solutions was the potential synergy between Rho-associated protein kinase (ROCK) 1/2 inhibitors and pan-HDACi (Table 1 and Appendix A). Interestingly, ROCK inhibition, besides its effects on invasion and tumor growth, was previously described as valuable “priming” strategy for chemotherapy of PDAC [33,34]. ROCK is also identified as one of the downstream effectors of the sphingolipid S1P signaling [21,35]. In the experimental analysis we have used ROCK1/2 inhibitor RKI-1447 (IC_50_ (ROCK1) = 14.5 nM, and IC_50_ (ROCK2) = 6.2 nM). Fingolimod—another compound predicted to be synergistic counterpart of pan-HDACi, was available for experimental evaluation in the laboratory at the time of the study. Fingolimod, the FDA-approved agonist and functional antagonist of S1P receptor subtype 1 (S1PR1) [36], was found to be a potent anticancer agent in some PDAC models which could be at least partially attributed to its moderate inhibitory effects of the class I of HDACs [37,38]. Besides S1P receptors, another elements of sphingolipid signaling pathway were recognized as off-targets of fingolimod, including ceramide synthase 2 (CERS2), S1P lyase, and SPHK1. Altogether, fingolimod appears to be an interesting chemical probe to tackle the potential of synergism between HDACi and sphingolipid signaling modulation [36]. However, it is important to note that fingolimod:HDACi pair was scored among first 13% of predictions indicating rather moderate, or less probable, synergy.

To assess the cytotoxic effects of the drug combinations, Chou-Talalay model [39] was used, which requires drugs to be administered at a fixed dose ratio. MIA PaCa-2 and Panc-1 cells were treated with a combination of synthesized HDACi and RKI-1447, or HDACi and fingolimod in a 4 × 4 dose matrix and measured the resultant cell viabilities by the MTT assay (Appendix A). Isobologram analysis of the drug combination treatment at low concentrations and concentrations up to the IC_50_ values revealed synergisms for a combination of compound 6b and RKI-1447 as well as compound 9b and RKI-1447 on MIA PaCa-2 cells (Table 3).

Synergistic effects of the combination treatment with compound 8b and RKI-1447 on cell viability were observed only at the highest tested concentration (Table 3 and Appendix A). Considering the results on Panc-1 cell line, isobologram analysis of the drug combination treatment revealed slight synergism for the combination of compound 9b and RKI-1447 only at the lowest tested concentrations (Table 3 and Appendix A). Similarly to the results on synergism between HDACi and RKI-1447, synergism between HDACi and fingolimod was also observed dominantly in 6b and 9b (Table 4 and Appendix A). 

In alignment with the predictions, synergisms between HDACi and fingolimod were moderate and observed mostly for the highest tested concentrations. Although the antagonisms were detected for some of the tested concentration pairs, these antagonisms are of less concern since the most of them were detected for lower fractions of the affected cell viability (third row of Table 3 and Table 4 of each pair).

It is important to note that HDAC inhibitors 6b, 8b and 9b were not part of the initial chemical sensitivity dataset used in bioinformatics analysis. Therefore, experimentally detected synergisms for 6b, 8b and 9b indicate that synergisms predicted by bioinformatics analysis are driven mostly by direct HDAC inhibition rather than off-target effects of specific HDAC inhibitors included in the chemical sensitivity dataset.

## 3. Materials and Methods

### 3.1. Data Preparation

Pharmaco-transcriptomics relationships between basal gene expression and chemical sensitivity of pancreatic carcinoma cell lines (CCLs) were investigated using systematic correlation analysis of large publically available data sets: Genomics of Drug Sensitivity in Cancer (GDSC, https://www.cancerrxgene.org/, accessed on 17 June 2022) [13] chemical sensitivity data sets, Cancer Therapeutics Response Portal (CTRP, https://portals.broadinstitute.org/ctrp.v2.1/, accessed on 17 June 2022) [11] chemical sensitivity data set, Cancer Cell Line Encyclopedia (CCLE, https://depmap.org/portal/download/ 22Q2, accessed on 17 June 2022) [14] basal gene expression data set. Pearson correlation coefficients have been calculated between area under the curve (AUC) values and basal expression data (as log2RSEM (RNA-Seq by Expectation Maximization)) for each transcript across all tested cell lines. Matrix of Pearson correlation coefficients was further normalized using Fisher’s z-transformation to adjust for variations in number of tested CCLs for each of analyzed molecules. Transcripts corresponding to the Tukey’s outlies (1.5 interquartile range) of each small molecule were further extracted and stored as a transcriptomic “fingerprints” of drug resistance or sensitivity.

### 3.2. Synergy Predictions

Each of the transcripts from sensitivity or resistance signatures was further screened on the significance of positive or negative correlation coefficients across all analyzed small molecules. Only significant correlations (*p*-value < 0.05) were counted and each iteration was recorded as “Count +/−“ or “Count −/+”. Total score, titled Final_Syn_Score, was calculated by summation across counts. List of annotated targets for all detected synergisms (Final_Syn_Score > 10 and at least one Tukey’s outlier transcript shared between small molecules) was further analyzed using the functional enrichment analysis from g:Profiler server (https://biit.cs.ut.ee/gprofiler, accessed on 5 September 2022) [40]. The Genotype-Tissue Expression (GTEx, https://www.gtexportal.org/home/, accessed on 20 September 2022) [41] database and The Cancer Genome Atlas (TCGA, https://www.cancer.gov/tcga, accessed on 20 September 2022) [42] were used to analyze the gene expression profiles of HDACs and elements of sphingolipid signaling across pancreatic carcinoma patient data. Data was accessed and analyzed through GEPIA web server (http://gepia.cancer-pku.cn/, accessed on 20 September 2022) [43]. Transcripts corresponding to the overlapped signatures identified across sphingolipid signaling pathway were extracted and analyzed through network analysis using the STRING [29] data in Cytoscape 3.9.1 [44]. To identify major contributors to the communication across networks and mechanism of synergy, BC analysis was performed using CytoNCA tool [45].

### 3.3. Cell Culture

MIA Paca-2 (ATCC CRL-1420), Panc-1 (ATCC CRL-1469) cells were cultured in DMEM medium (Sigma-Aldrich, St. Louis, MO, USA) supplemented with 10% fetal bovine serum (FBS) (Sigma-Aldrich, St. Louis, MO, USA), 100 µg/mL streptomycin and 100 UmL-1 penicillin (Sigma-Aldrich, St. Louis, MO, USA), and grown as a monolayer in humidified atmosphere of 95% air and 5% CO_2_ at 37 °C. in a 5% CO_2_ atmosphere at 370C and in humidified incubator.

### 3.4. Quantitative Analysis of Drug Synergy

To determine the synergistic effects of the drug combinations, we performed an MTT viability assay and the combination index method described by Chou and Talalay [39]. Cytotoxic activity of synthesized HDAC inhibitors, ROCK inhibitor RKI-1447 (Selleck Co. (Shanghai, China) and Fingolimod HCl (Sigma-Aldrich, St. Louis, MO, USA) was assessed on MIA PaCa-2 and Panc-1 cells using MTT assay [46]. MIA PaCa-2 (4 × 103 cells/well) and PANC-1 (5 × 103 cells/well) were treated with synthesized compounds in five different concentrations (100, 50, 25, 12.5, and 6.25 μM), and each concentration is added in five replicates. After 72 h, 20 µL of MTT solution (3-(4, 5-dimethylthiazol-2-yl)-2, 5-dyphenyl tetrazolium bromide) (Sigma-Aldrich, St. Louis, MO, USA) was added to each well. Samples were incubated for 4 h, followed by the addition of 100 μL of 10% SDS and incubated at 37 °C. Absorbance at 570 nm was measured the next day. Cell survival (%) was calculated as an absorbance (570 nm) ratio between treated and control cells multiplied by 100. IC50 was defined as the concentration of the agent that inhibited cell survival by 50% compared to the vehicle control.

To determine the synergistic effects of the drug combinations, we performed the combination index method described by Chou and Talalay [39], using the CalcuSyn software (version 2.0 Biosoft, Cambridge, UK). The combination index (CI) was calculated to assess the benefits of combinational treatment in MiaPaCa-2 and PANC-1 cells. A CI equal to 1 indicates an additive effect, CI = 0.3–0.7 indicates synergism, CI = 0.1–0.3 indicates strong synergism, and CI < 0.1 indicates very strong synergism.

## 4. Conclusions

Pancreatic ductal adenocarcinoma (PDAC) is considered as one of the most aggressive and lethal malignancies. Due to the high chemoresistance of PDAC as well as difficult early diagnosis which hampers surgical resection, there is an urgent need to develop novel pharmacological approaches for effective treatment of PDAC. In an emerging era of personalized medicine, systematically tailored combination therapy holds the promise for advancing treatment of PDAC, but number of possible combinations considering all approved drugs and drug candidates in clinical trials is too large to be explored experimentally. In respect to the many epigenetic alterations in PDAC, epigenetic drugs including HDAC inhibitors could be seen as game changers. In this work, we explored possibility of using drug-sensitivity data together with basal gene expression data on pancreatic cell lines to predict the combinatorial options available for HDACi and developed bioinformatics screening protocol for predictions of synergy in PDAC. Our results identified sphingolipid signaling pathway with associated downstream effectors as a promising novel target for future development of multi-target therapeutics or combined therapy with HDACi. The experimental validation of the protocol led to the discovery of the novel HDACi-ROCK inhibitor synergism as well as HDACi-fingolimod synergism. Further experimental evaluation of identified synergism between modulation of sphingolipid signaling and HDAC inhibition will be the major direction of our future work. All predictions made through this study are freely available as a part of Appendix A.

## Figures and Tables

**Figure 1 pharmaceuticals-16-00294-f001:**
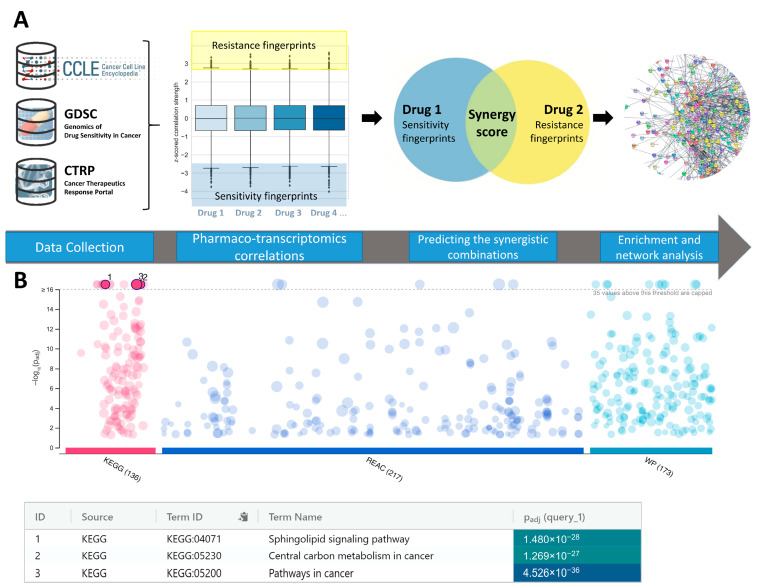
Overall design of the study and results of enrichment analysis. (**A**) The preview of the approach used to identify synergistic interactors of HDACi; (**B**) Results of the pathway enrichment analysis indicated sphingolipid signaling pathway as a novel target for synergistic treatment of PDAC.

**Figure 2 pharmaceuticals-16-00294-f002:**
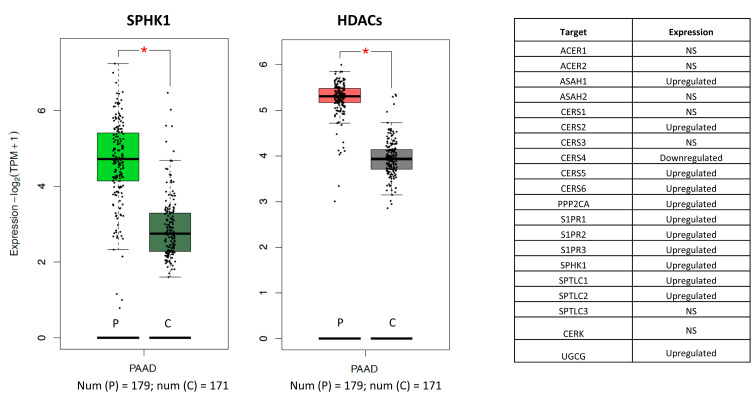
SPHK1 and HDACs (class I) are upregulated in pancreatic cancer. Left—Box plot of the expression for the central regulator of sphingolipid rheostat; Middle—Box plot of the expression of class I HDACs; Right—Table of summarized expressions for all component of sphingolipid signaling pathway identified in the study. P indicates patient tissue, C indicates healthy control, the asterisk (*) indicates significant change in the expression (*p*-value cutoff 0.01) while NS indicates non-significant change in expression between patient and normal tissue.

**Table 1 pharmaceuticals-16-00294-t001:** Results of HDACi synergy predictions with known literature data.

Molecule 1 (HDACi)	Molecule 2	Target Profile of Molecule 1	Target Profile of Molecule 2	Count +/− *^a^*	Count −/+ *^b^*	Final_Syn_Score *^c^*
tacedinaline	RG-108	HDAC1; HDAC2; HDAC3; HDAC6; HDAC8	DNMT1	37	403	440 (0.3%)
Merck60	Sorafenib	HDAC1; HDAC2	BRAF; FLT3; KDR; RAF1	14	61	75 (5%)
tacedinaline	PI-103	HDAC1; HDAC2; HDAC3; HDAC6; HDAC8	PI3Kalpha, DAPK3, CLK4, PIM3, HIPK2	23	243	266 (1%)
tacedinaline	KU-0063794	HDAC1; HDAC2; HDAC3; HDAC6; HDAC8	MTOR	189	425	614 (0.1%)
BRD-K85133207	AZD5153	HDAC1	BRD4	36	47	83 (4.6%)
PCI-34051	gemcitabine	HDAC8, HDAC6, HDAC1	CMPK1; RRM1; TYMS	32	34	66 (5.7%)
PCI-34051	GSK429286A	HDAC8, HDAC6, HDAC1	ROCK1, ROCK2	65	79	144 (2.3%)
BRD-K51490254	SKI-II	HDAC6; HDAC8	SPHK1	12	63	75 (5%)
PCI-34051	fingolimod	HDAC8, HDAC6, HDAC1	S1PR1	10	17	27 (12.7%)

*^a^* Count of positively correlated (Molecule 1) vs. negatively correlated (Molecule 2) transcripts; *^b^* Count of negatively correlated (Molecule 1) vs. positively correlated (Molecule 2) transcripts. *^c^* The numbers in brackets indicates position of the solution in the ranked list.

**Table 2 pharmaceuticals-16-00294-t002:** Communication hubs identified in network of transcripts involved in predicted synergism between HDACi and sphingolipid signaling pathway.

Gene ID	BC	Gene ID	BC
TP53	56,527	APOA1	8858
VWF	19,076	ETV1	8308
CALR	17,531	PEBP1	7894
NTRK1	16,901	ITGA4	7747
MMP9	16,341	GNGT2	7426
CYCS	13,336	SLC17A7	7102
DNM1	11,804	TPT1	6997
FOXP3	10,884	BUD13	6632
H2AFX	10,415	COPS5	6290
RCC1	9091	DDB2	6156

**Table 3 pharmaceuticals-16-00294-t003:** Synergism analysis for combination of novel HDACi and RKI-1447.

	MIA PaCa-2 Cell Line				Panc-1 Cell Line			
	Compound Concentration *^a^*				Compound Concentration *^a^*			
	I	II	III	IV	I	II	III	IV
HDACi (6b)	48.60 ± 7.45	13.58 ± 2.33	4.80 ± 1.01	2.91 ± 0.87	23.61 ± 3.26	1.39 ± 0.55	0.47 ± 0.21	0.5 ± 0.2
ROCKi (RKI-1447)	64.18 ± 8.12	40.49 ± 5.55	37.49 ± 3.02	30.84 ± 7.33	62.76 ± 11.23	33.89 ± 8.92	22.02 ± 7.22	18.11 ± 5.90
6b + RKI-1447	83.28 ± 10.34	55.48 ± 7.51	37.69 ± 5.38	38.31 ± 4.56	69.79 ± 10.45	48.51 ± 6.34	24.98 ± 3.13	8.92 ± 4.21
Combination Index (CI)	0.445	0.981	1.253	0.605	0.992	1.050	1.334	2.117
Interaction 6b + RKI-1447 *^b^*	+ + +	+	− −	+ +	±	±	− −	− − −
HDACi (8b)	27.30 ± 4.44	1.83 ± 0.70	0.94 ± 0.45	0.57 ± 0.19	49.78 ± 6.77	38.21 ± 5.89	17.13 ± 2.11	0.1 ± 0.05
ROCKi (RKI-1447)	62.94 ± 10.11	42.93 ± 6.99	36.07 ± 7.51	35.46 ± 4.09	58.78 ± 5.55	30.77 ± 3.45	17.27 ± 2.28	7.41 ± 2.56
8b + RKI-1447	73.02 ± 7.44	45.32 ± 5.14	31.14 ± 5.21	28.91 ± 2.34	63.94 ± 6.33	42.96 ± 5.79	21.99 ± 3.45	6.79 ± 3.06
Combination Index (CI)	0.333	1.564	2.469	1.505	3.133	1.433	1.266	1.474
Interaction 8b + RKI-1447 *^b^*	+ + +	− − −	− − −	− − −	− − −	− −	− −	− − −
HDACi (9b)	44.65 ± 3.39	18.25 ± 2.22	1.71 ± 0.67	0.46 ± 0.20	45.18 ± 4.55	23.14 ± 7.42	7.56 ± 3.71	1.12 ± 0.89
ROCKi (RKI-1447)	61.64 ± 7.78	43.28 ± 5.38	35.02 ± 3.25	29.16 ± 2.16	64.82 ± 6.41	37.92 ± 5.34	19.76 ± 2.30	9.41 ± 1.06
9b + RKI-1447	78.03 ± 7.18	64.37 ± 6.78	41.91 ± 5.98	33.32 ± 4.21	64.68 ± 5.01	51.26 ± 7.20	33.14 ± 6.10	18.01 ± 4.02
Combination Index (CI)	0.513	0.577	0.964	0.803	1.705	1.207	0.975	0.823
Interaction 9b + RKI-1447 *^b^*	+ + +	+ + +	±	+ +	− − −	− −	±	+

*^a^* Concentrations are presented in descending order (I is the highest, and IV the lowest tested concentration). Exact concentrations for each compound are presented in Appendix A. *^b^* Legend: (+ + +) Synergism; (+ +) Moderate synergism; (+) Slight synergism; (±) Nearly additive; (− −) Moderate antagonism; (− − −) Antagonism.

**Table 4 pharmaceuticals-16-00294-t004:** Synergism analysis for combination of novel HDACi and fingolimod.

	MIA PaCa-2 Cell Line				Panc-1 Cell Line			
	Compound Concentration *^a^*				Compound Concentration *^a^*			
	I	II	III	IV	I	II	III	IV
HDACi (6b)	71.49 ± 11.91	41.62 ± 11.91	23.06 ± 9.75	16.27 ± 5.34	63.68 ± 4.70	25.29 ± 6.71	23.93 ± 6.23	10.30 ± 1.96
Fingolimod	79.98 ± 10.36	48.31 ± 12.04	20.57 ± 4.85	20.33 ± 5.37	79.18 ± 1.45	40.09 ± 4.97	24.56 ± 1.24	18.99 ± 0.95
6b + Fingolimod	89.28 ± 4.05	60.81 ± 20.19	34.11 ± 14.58	26.96 ± 7.59	84.96 ± 2.05	67.22 ± 5.98	37.87 ± 5.71	21.98 ± 0.78
Combination Index (CI)	0.780	1.422	1.648	1.070	0.847	0.945	1.232	1.141
Interaction 6b + Fingolimod *^b^*	+ +	− −	− − −	±	+ +	±	− −	−
HDACi (8b)	81.22 ± 0.96	68.72 ± 2.69	36.08 ± 3.29	16.72 ± 4.28	58.10 ± 0.24	23.86 ± 9.24	4.77 ± 2.69	4.68 ± 2.33
Fingolimod	87.22 ± 2.01	52.02 ± 2.47	23.73 ± 3.92	22.83 ± 1.89	68.24 ± 4.62	41.81 ± 5.37	19.44 ± 10.97	12.27 ± 8.75
8b + Fingolimod	91.31 ± 10.08	65.06 ± 1.92	38.98 ± 5.26	26.55 ± 2.94	74.74 ± 5.61	30.91 ± 6.84	16.60 ± 7.22	9.42 ± 5.57
Combination Index (CI)	1.188	1.836	1.852	1.338	1.267	1.806	2.027	1.592
Interaction 8b + Fingolimod *^b^*	− − −	− − −	− − −	− −	− −	− − −	− − − −	− − −
HDACi (9b)	83.61 ± 0.27	69.42 ± 11.34	53.31 ± 17.80	44.84 ± 11.21	61.28 ± 6.91	50.93 ± 3.83	40.71 ± 3.74	24.03 ± 5.73
Fingolimod	76.58 ± 4.48	63.82 ± 6.81	43.78 ± 12.65	35.34 ± 6.99	62.10 ± 4.94	50.04 ± 3.20	31.21 ± 11.85	21.17 ± 3.33
9b + Fingolimod	89.11 ± 1.57	73.59 ± 12.40	55.57 ± 18.16	47.22 ± 9.51	76.50 ± 5.14	66.12 ± 10.76	53.41 ± 18.12	33.34 ± 10.07
Combination Index (CI)	0.870	1.449	1.773	1.293	0.911	0.848	0.810	1.120
Interaction 9b + Fingolimod *^b^*	+	− −	− − −	− −	±	+ +	+ +	−

*^a^* Concentrations are presented in descending order (I is the highest, and IV the lowest tested concentration). Exact concentrations for each compound are presented in Appendix A. *^b^* Legend: (+ +) Moderate synergism; (+) Slight synergism; (±) Nearly additive; (−) Slight antagonism; (− −) Moderate antagonism; (− − −) Antagonism; (− − − −) Strong antagonism.

## Data Availability

Data is contained within the article and Appendix A.

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
