# Peer review of "Correlating Basal Gene Expression across Chemical Sensitivity Data to Screen for Novel Synergistic Interactors of HDAC Inhibitors in Pancreatic Carcinoma"

_pharmaceuticals, 2023, doi:10.3390/ph16020294_

Round 1

Reviewer 1 Report

Authors developed approach for utilising drug-sensitivity data together with basal gene expression of pancreatic cell lines to predict combinatorial options available for histone deacetylase inhibitors (HDACi). Furthermore, authors made all predictions freely available what is valuable contribution for future studies.  

Study is well designed and explained. Regarding the used methods a few clarifications are needed:

·        Authors considered only transcripts with significant correlation, what was the threshold for counting correlation as significant?

·        Why authors have chosen betweenness centrality measure to identify the most important communication hubs over other network centrality measures (e.g eigenvector centrality, closeness centrality..)?

·        Chou-Talalay model should be cited also at the first place where it is mentioned (line 262)

Subsection 3.1. Computational procedure would be easier to follow if divided into two parts: Data and Computational procedure

Reviewer 2 Report

In the present manuscript titled “Correlating basal gene expression across chemical sensitivity data to screen for novel synergistic interactors of HDAC inhibitors in pancreatic carcinoma,” authors have made good effort of utilizing already available data to predict better treatment regime for pancreatic patients. The method utilized for relating the differential gene expression data of pancreatic cancer cell lines with the patterns of small molecule sensitivity to predict combinatorial options available for HDACi is good. It will be good if following suggestions or queries are also included in the manuscript.

1.      In the results section authors have mentioned “Considering the fact that we were not tested exact predicted pairs of 282 small-molecules, experimentally detected synergisms could indicate that these synergisms are driven by on-target rather than off-target effects of these drugs.” Is not clear.

2.      It will be good if they would include a panel of cancer cell lines in their validation experiments.

3.      In the results section, it will be good if authors can modify certain sentences to make it clearer and crisper to the readers.

4.      The quality of figures is poor.

5.      There should be proper figure legends for figures in supplementary data too because it will be more understandable.
